# A Nonlinear Free Vibration Analysis of Functionally Graded Beams Using a Mixed Finite Element Method and a Comparative Artificial Neural Network

**Chih-Ping Wu *** , **Shu-Ting Yeh and Jia-Hua Liu**

Department of Civil Engineering, National Cheng Kung University, Tainan 70101, Taiwan;
sky148345240@gmail.com (S.-T.Y.); abirritate1318@gmail.com (J.-H.L.)
*  Correspondence: cpwu@mail.ncku.edu.tw

**Abstract:** Based on the Hamilton principle combined with the Timoshenko beam theory, the authors developed a mixed finite element (FE) method for the nonlinear free vibration analysis of functionally graded (FG) beams under combinations of simply supported, free, and clamped edge conditions. The material properties of the FG beam gradually and smoothly varied through the thickness direction according to the power-law distributions of the volume fractions in the constituents, and the effective material properties of the FG beam were estimated using the rule of mixtures. The von Kármán geometrical nonlinearity was considered. The FE solutions of the amplitude-frequency relations of the FG beam were obtained using an iterative process. Implementing the mixed FE method showed that its solutions converged rapidly and that the convergent solutions closely agreed with the accurate solutions reported in the literature. A multilayer perceptron (MP) back propagation neural network (BPNN) was also developed to predict the nonlinear free vibration behavior of the FG beam. After appropriate training, the prediction of the MP BPNN's amplitude-frequency relations was entirely accurate compared to those obtained using the mixed FE method, and its central processing unit time was less time-consuming than that of the mixed FE method.

**Keywords:** artificial neural networks; finite element methods; functionally graded beams; Hamilton's principle; mixed Timoshenko beam theory; nonlinear vibration

## 1. Introduction

In recent years, functionally graded (FG) material, which is composed of a two- or multi-phase material according to some spatial distributions of the volume fractions of these constituents, has gradually become an important industrial material and has been used to form a variety of beam-, plate- and shell-like structures in advanced engineering due to the flexibility of its application and the continuous distributions of its material properties over the physical domain of the structures in which it is used [1,2]. FG structures could be designed to improve their specific structural performances by changing the volume fractions of the constituents. In addition, FG structures could replace conventional fiber-reinforced composite (FRC) structures to prevent delamination failure, which often occurs at the interfaces between the adjacent layers of FRC structures due to the material properties that suddenly change at such places. Some review articles discussing the development, manufacture, and application of FG and FRC structures and their corresponding mechanical analyses can be found in the literature [3,4]. Among these, the review conducted in this work focuses on the literature related to the linear and nonlinear free vibration analyses of FG beams with different boundary conditions.

The linear structural analyses of FG beams have been presented. Based on the principle of virtual displacements (PVD) combined with the first-order shear deformation theory (FSDT), Chakraborty et al. [5,6] developed a new beam element for the static bending, thermo-elastic, free vibration, and wave propagation analyses of FG beams. The differences

between the above-mentioned structural behaviors of the metal-ceramic two-phase FG beam and those of the pure metal and ceramic beams were highlighted. Aydogdu and Taskin [7] studied the free vibration responses of FGM beams using various equivalent single-layered theories (ESLTs), such as the classical beam theory, the FSDT, and the higher-order shear deformation theory (HSDT). Simsek [8,9] investigated the static bending, free vibration behaviors of FG beams under various boundary conditions using the HSDT, in which some effects on the deflection and frequency parameters of the FG beams were examined, such as the effects of slender ratios, material-property gradient indices, and different formulations. The HSDT was extended to the thermo-mechanical vibration analysis of sandwich beams with FG carbon nanotube-reinforced composite (CNTRC) face sheets by Ebrahimi and Farazmandnia [10]. Using Carrera's unified formulation (CUF) [11], Pietro et al. [12] developed a hierarchical one-dimensional finite element method to study the thermo-elastic response of FG beams. Coskun et al. [13] developed a third-order plate theory for the static bending, free vibration, and static buckling analyses of FG porous microplates, where three different porosity distributions through the thickness direction of the plate were considered. Tham et al. [14] investigated the free vibration behavior of laminated FG CNTRC doubly curved shallow shells using a new four-variable refined theory. Wu and Xu [15] presented strong and weak formulations of a mixed higher-order formulation for the static bending analysis of FG beams when subjected to thermo-mechanical loads. Based on the mixed higher-order shear deformation theory, Wu and Li [16] examined the thermal buckling behavior of FG beams, and they also developed the multi-objective optimization of FG beams to maximize critical temperature change parameters and minimize the total mass of the FG beam using a non-dominated sorting-based genetic algorithm.

ESLTs combined with the von Kármán geometrical nonlinearity (VKGN) were also applied as extensions to FG beams' geometrical nonlinear static and free vibration analyses. Ma and Lee [17] presented an exact, closed-form solution for the nonlinear static responses of single-layered FG beams under different boundary conditions using the FSDT and accounting for the VKGN. Based on the HSDT and using a physically neutral surface and the VKGN, Zhang [18] examined the nonlinear bending behavior of single-layered FG beams using the Ritz method. The HSDT was also extended to a bending analysis of sandwich beams with CNTRC face sheets by Salami [19] and a nonlinear bending analysis of FG CNTRC plates in thermal environments by Shen [20]. Based on the FSDT, Ghayesh [21] investigated the nonlinear forced vibration behavior of axially FG Timoshenko tapered beams. Shen et al. [22] presented the nonlinear vibration analysis of FG graphene-reinforced composite laminated beams when resting on elastic foundations in thermal environments using the HSDT. Ding et al. [23] applied the Euler-Bernoulli beam theory to the nonlinear vibration analysis of FG beams, in which the effects of the rotary inertia of the cross-section and the neural surface position were considered. Eltaher et al. [24] carried out the post-buckling and nonlinear vibration analyses of beams resting on a nonlinear foundation. Based on Hamilton's principle, Chaudhari and Lal [25] carried out a nonlinear free vibration analysis of FG CNTRC beams, which were subjected to thermal loading, in which the HSDT and the VKGN were used to derive a weak-form formulation, from which a Lagrangian $C^0$ element with four degrees-of-freedom per node was developed. Based on the FSDT and combined with the VKGN, Mirzaei and Kiani [26] analyzed the nonlinear free vibration characteristics of temperature-dependent sandwich beams with CNTRC face sheets. Based on the third-order shear deformation theory, Babaei et al. [27] investigated the small- and large-amplitude-free vibration behaviors of FG beams resting on an elastic foundation consisting of Winkler, shear, and nonlinear springs.

Over the past few decades, artificial neural networks (ANNs) have emerged as useful mathematical tools with a self-learning ability and have been used in various advanced engineering applications. The ANN approach was inspired by the biological neural system, such that its algorithm was designed as a multilayer structure in which each layer consists of some interconnected neurons. The ANN was trained using some given data sources,

and the learning experience was stored in the weight numbers and biases, which were used to connect the relations between the outputs of the last layer and the inputs of each neuron in the current layer. Due to its superior abilities to handle massively parallel computation and self-learning, the ANN can be conventionally used in function approximations, pattern recognition, classification, etc. [28]. Recently, ANNs have further shown their diverse applications, such as machine learning [29], the modeling of structural and material behavior [30], and optimal design [31,32].

Chakraverty et al. [33] developed an ANN to estimate the vibration characteristics of plate structures, in which initial weight numbers were generated using regression analysis. Reddy et al. [34] used an ANN to predict the natural frequencies of laminated composite plates under clamped boundary conditions, in which a multilayer perceptron (MP) back propagation neural network (BPNN) was used, and the data sources obtained using the linear shell elements were provided for training. Jodaei et al. [35] developed a state space differential quadrature (SSDQ) method for the free vibration analysis of functionally graded (FG) annular plates under various boundary conditions, and these SSDQ solutions were used to train an ANN. Fetene et al. [36] developed a FEM-based neural network for the inverse prediction of bending a cantilever beam. The comparative modeling of the static and buckling behavior of laminated composite structures with the aid of ANNs has also been proposed by Subramani and Sharmila [37] and Liu et al. [38], respectively.

From the above literature survey, the authors found that the analyses mentioned above were rarely based on the weak-form formulation compared with those based on the strong-form formulation, and the corresponding equilibrium and motion equations derived using the Reissner mixed variation theorem (RMVT) were even fewer than those derived using the PVD, even though RMVT-based models were concluded to be superior to PVD-based models for the various analyses of plates and shells [11]. Hence, developing an RMVT-based weak-form formulation for analyzing FG beams is important in academic research and practical applications. In addition, as mentioned above, ANNs can be used to predict the structural behavior of FG beams by feeding them suitable data sources; thus, the development of a competitive ANN algorithm is necessary for future advanced studies with a lot of time-consuming characteristics, such as nonlinear modeling, optimal design, system recognition, etc.

In this work, the authors derived a weak-form formulation of an RMVT-based Timoshenko beam theory (TBT) for the large-amplitude free vibration analysis of FG beams under combinations of simply supported, free, and clamped edge conditions, in which the VKGN effect was considered. Nonlinear equilibrium equations of the finite element (FE) method based on the mixed TBT were derived using a variational approach, and an iterative process was used to obtain the numerical solutions for the issue of interest. The material properties were assumed to obey the power-law distributions that varied through the thickness direction of the FG beam according to the volume fractions of the constituents, and the effective material properties were estimated using the rules of mixtures. Four different boundary conditions, simple-simple (S-S), clamped-clamped (C-C), clamped-simple (C-S), and clamped-free (C-F), were considered. A parametric study of the critical effects on the amplitude-frequency relations in significant amplitude-free vibration cases was conducted, including geometrical nonlinearity, boundary conditions, aspect ratios, and material-property gradient indices. In addition, an MP BPNN was developed to predict the amplitude-frequency relations of FG beams, in which the Levenberg–Marquardt algorithm [28] was adopted to speed up the convergence rate of the MP BPNN. Some critical effects on the performance of the MP BPNN were closely examined, such as the number of layers used in the ANN and the number of neurons in each layer.

## 2. Weak-Form Formulation of the Nonlinear Mixed TBT

Based on the mixed TBT, a weak-form formulation for a large amplitude free vibration analysis of moderately thick FG rectangular beams under various boundary conditions was developed in this section. The symbols $L$, $h$, and $b$ denote the FG beam's length, thickness,

and width. In addition, a set of Cartesian coordinates $(x, y, z)$ for the kinematics description of the FG beam was located at the mid-surface of its left end.

The structural behavior of the FG beam was described using a mixed TBT [39–42], in which the shear deformation effects were considered to be a constant through the thickness direction of the FG beam, and the related displacement field was given, as follows:

$$u_x(x, z) = u(x) - z\phi(x), \tag{1}$$

$$u_y(x, z) = 0, \tag{2}$$

$$u_z(x, z) = w(x), \tag{3}$$

where $u_i(x, z)$ $(i = x, y,$ and $z)$ denote the displacement components of the FG beam in the $x, y,$ and $z$ directions, respectively. $u(x)$ and $w(x)$ stand for the mid-plane displacement components of the FG beam in the $x$ and $z$ directions and $\phi(x)$ is the total rotation in the $x$–$z$ plane.

The strain-displacement relations of the FG beam considering the VKGN effect were given by:

$$\varepsilon_x = u_{,x} - z\,\phi_{,x} + (1/2)(w_{,x})^2, \tag{4}$$

$$\gamma_{xz} = -\phi + w_{,x}, \tag{5}$$

$$\varepsilon_y = \varepsilon_z = \gamma_{yz} = \gamma_{xy} = 0, \tag{6}$$

where $\varepsilon_x$, $\varepsilon_y$, $\varepsilon_z$, $\gamma_{xz}$, $\gamma_{yz}$ and $\gamma_{xy}$ are the strain components of the FG beam, and $g_{,x} = \partial g/\partial x$, in which $g = u,\ w,\ $ and $\phi$.

The nonzero stress components of the FG beam were given by:

$$\sigma_x = E\left[u_{,x} - z\,\phi_{,x} + (1/2)(w_{,x})^2\right], \tag{7}$$

$$\tau_{xz} = k_c\,G(-\phi + w_{,x}), \tag{8}$$

where $G$ and $E$ are defined as the shear and Young's moduli of the FG beam, respectively, and in general, these are specific functions of $z$, i.e., $G = G(z)$ and $E = E(z)$. $k_c$ denotes the shear stress correction factor of the FG beam, which was taken as 5/6 in this work.

The generalized force resultants, namely the axial force $N$, moment $M$, and shear force $Q$, of the FG beam, were defined by:

$$N = A_{xx}\,u_{,x} - B_{xx}\,\phi_{,x} + (A_{xx}/2)(w_{,x})^2, \tag{9}$$

$$M = -B_{xx}\,u_{,x} + D_{xx}\,\phi_{,x} - (B_{xx}/2)(w_{,x})^2, \tag{10}$$

$$Q = -k_c\,A_{xz}(-\phi + w_{,x}), \tag{11}$$

where for an $N_l$ − layered FG beam:

$$A_{xx} = \sum_{m=1}^{N_l} b \int_{z_{m-1}}^{z_m} E(z)\,dz,\quad B_{xx} = \sum_{m=1}^{N_l} b \int_{z_{m-1}}^{z_m} E(z)\,z\,dz,\quad D_{xx} = \sum_{m=1}^{N_l} b \int_{z_{m-1}}^{z_m} E(z)\,z^2\,dz,\quad A_{xz} = \sum_{m=1}^{N_l} b \int_{z_{m-1}}^{z_m} G(z)\,dz,$$

where $N_l$ denotes the total number of layers constituting the multi-layered beam, and $z_m$ and $z_{m-1}$ are the thickness coordinates of the top and bottom surfaces of the $m^{\text{th}}$-layer when measured from the mid-surface of the FG beam.

Four different boundary conditions of the FG beam were considered as follows:

Case 1. S-S supported:

$$u = w = M = 0 \quad \text{at } x = 0 \text{ and } x = L; \tag{12}$$

Case 2. C-S supported:

$$
\begin{aligned}
u = w = \phi = 0 \quad &\text{at } x = 0, \\
\text{and } u = w = M = 0 \quad &\text{at } x = L;
\end{aligned} \tag{13}
$$

Case 3. C-C supported:

$$u = w = \phi = 0 \quad \text{at } x = 0 \text{ and } x = L; \tag{14}$$

Case 4. C-F supported:

$$
\begin{aligned}
u = w = \phi = 0 \quad &\text{at } x = 0, \\
\text{and } N = M = Q = 0 \quad &\text{at } x = L.
\end{aligned} \tag{15}
$$

Based on Hamilton's principle, a weak-form formulation was derived using a variational approach, in which the RMVT combined with the TBT and VKGN kinematics was used. The energy functional of the FG beam could be written in the form of:

$$I = \int_{t_1}^{t_2} (T - \Pi_R)\, dt, \tag{16}$$

where $T$ and $\Pi_R$ represent the kinetic energy and Reissner's potential energy of the FG beam, respectively, and can be given by:

$$
\begin{aligned}
T = \sum_{e=1}^{N_e} \int_{x_e}^{x_{e+1}} &\left[ (1/2)m_0 \left( \partial u^{(e)}/\partial t \right)^2 + (1/2)m_2 \left( \partial \phi^{(e)}/\partial t \right)^2 \right. \\
&\left. + (1/2)m_0 \left( \partial w^{(e)}/\partial t \right)^2 - m_1 \left( \partial u^{(e)}/\partial t \right)\left( \partial \phi^{(e)}/\partial t \right) \right] dx,
\end{aligned} \tag{17}
$$

$$
\begin{aligned}
\Pi_R = \sum_{e=1}^{N_e} \int_{x_e}^{x_{e+1}} &\left\{ N^{(e)} \left[ u^{(e)}{}_{,x} + (1/2)\left( w^{(e)}{}_{,x} \right)^2 \right] + M^{(e)} \left( \phi^{(e)}{}_{,x} \right) + Q^{(e)} \left( \phi^{(e)} - w^{(e)}{}_{,x} \right) - (S_{Axx}/2)\left( N^{(e)} \right)^2 - (S_{Dxx}/2)\left( M^{(e)} \right)^2 \right. \\
&\left. - S_{Bxx}\left( N^{(e)} \right)\left( M^{(e)} \right) - [1/(2k_c A_{xz})]\left( Q^{(e)} \right)^2 \right\} dx + \overline{N}_0\, u_0 - \overline{N}_L\, u_L + \overline{M}_0 \phi_0 - \overline{M}_L \phi_L - \overline{V}_0 w_0 + \overline{V}_L w_L,
\end{aligned} \tag{18}
$$

where the superscript $e$ denotes a typical beam element, $N_e$ denotes the total number of beam elements constituting the FG beam; $m_0$, $m_1$, and $m_2$ are the mass of the cross-sectional area and its moment of inertia, and are given by $m_0 = \int_A \rho\, dA$, $m_1 = \int_A \rho\, z\, dA$, and $m_2 = \int_A \rho\, z^2\, dA$; $\delta_{kl}$ is the Kronecker delta symbol, in which $\delta_{kl} = 1$ when $k = l$, while $\delta_{kl} = 0$ when $k \neq l$; $S_{Axx} = D_{xx}/(A_{xx}D_{xx} - B_{xx}B_{xx})$, $S_{Bxx} = B_{xx}/(A_{xx}D_{xx} - B_{xx}B_{xx})$, and $S_{Dxx} = A_{xx}/(A_{xx}D_{xx} - B_{xx}B_{xx})$; when the material properties constituting the FG beam were symmetric with respect to the mid-surface, $B_{xx} = 0$; otherwise, $B_{xx} \neq 0$. In addition, the variables, $\overline{N}_0$, $\overline{N}_L$, $\overline{M}_0$, $\overline{M}_L$, $\overline{V}_0$, and $\overline{V}_L$, represented the applied axial forces, moments, and shear forces of the FG beam at the edges.

Based on the RMVT-based TBT, the generalized displacement variables $u^{(e)}$, $w^{(e)}$, and $\phi^{(e)}$ and the generalized force resultant variables $N^{(e)}$, $M^{(e)}$, and $Q^{(e)}$ of each element were selected as the primary variables subject to variation, and the time function of each field variable was assumed to be the harmonic function, i.e., $F^{(e)}(x, z, t) = \widetilde{F}^{(e)}(x, z)\, e^{i\,\omega\, t}$, in which $t$ represents the time variable, and $\omega$ is the natural frequency of the FG beam.

Then, by applying Hamilton's principle and imposing the continuity conditions at the nodes of the beam element, the authors finally obtained the motion equations of the loaded FG beam, as follows:

$$
\sum_{e=1}^{N_e}\left(\begin{bmatrix} 0 & 0 & 0 & k_{14}^{(e)} & 0 & 0 \\ 0 & 0 & 0 & 0 & 0 & k_{26}^{(e)} \\ 0 & 0 & 0 & 0 & k_{35}^{(e)} & k_{36}^{(e)} \\ k_{41}^{(e)} & 0 & 0 & k_{44}^{(e)} & k_{45}^{(e)} & 0 \\ 0 & 0 & 0 & k_{54}^{(e)} & k_{55}^{(e)} & 0 \\ 0 & k_{62}^{(e)} & k_{63}^{(e)} & 0 & 0 & k_{66}^{(e)} \end{bmatrix}^{(m)} + \begin{bmatrix} 0 & 0 & 0 & 0 & 0 & 0 \\ 0 & g_{22}^{(e)} & 0 & 0 & 0 & 0 \\ 0 & 0 & 0 & 0 & 0 & 0 \\ 0 & g_{42}^{(e)} & 0 & 0 & 0 & 0 \\ 0 & 0 & 0 & 0 & 0 & 0 \\ 0 & 0 & 0 & 0 & 0 & 0 \end{bmatrix}^{(m-1)} - \omega^2 \begin{bmatrix} m_{11}^{(e)} & 0 & m_{13}^{(e)} & 0 & 0 & 0 \\ 0 & m_{22}^{(e)} & 0 & 0 & 0 & 0 \\ m_{31}^{(e)} & 0 & m_{33}^{(e)} & 0 & 0 & 0 \\ 0 & 0 & 0 & 0 & 0 & 0 \\ 0 & 0 & 0 & 0 & 0 & 0 \\ 0 & 0 & 0 & 0 & 0 & 0 \end{bmatrix}^{(m)} \begin{Bmatrix} u_j^{(e)} \\ w_j^{(e)} \\ \phi_j^{(e)} \\ N_j^{(e)} \\ M_j^{(e)} \\ Q_j^{(e)} \end{Bmatrix}_{i=j=1,\dots,\,n_d}^{(m)} = \sum_{e=1}^{N_e}\begin{Bmatrix} 0 \\ 0 \\ 0 \\ 0 \\ 0 \\ 0 \end{Bmatrix}_{i=1,\dots,\,n_d}^{(m)}, \quad (19)
$$

where $k_{ij}^{(e)}$, $g_{ij}^{(e)}$ and $m_{ij}^{(e)}$ are relevant coefficients and the superscripts $m$ and $(m-1)$ refer to the $m^{\text{th}}$ and $(m-1)^{\text{th}}$ iterative processes.

Based on the linear vibration theory without considering the VKGN, the linear natural frequencies of the FE beam, which were independent of the modal variables, could be obtained by letting the geometrical stiffness coefficients be zero (i.e., $g_{ij}^{(e)} = 0$). Then, the corresponding eigenvectors were scaled up by allowing the maximum transverse displacement of the FG beam to be identical to a given vibration amplitude $\overline{w}_{\text{max}}$, and $g_{ij}^{(e)}$ could be determined using these scaled-up eigenvectors. The nonlinear natural frequencies of the FG beam, which were dependent upon the modal variables, could thus be obtained using the updated eigenvalue system equations (Equation (19)). The iteration process ended when the relative error between the nonlinear frequencies of $m^{\text{th}}$ and $(m-1)^{\text{th}}$ iterations were less than $10^{-5}$.

Using Equation (19) and the above-mentioned iterative process, the authors could investigate the large amplitude free vibration characteristics of moderately thick FG beams under different boundary conditions.

## 3. The MP BPNN

### 3.1. Feeding Forward Process

An MP BPNN was developed and briefly described in this section. For illustrative purposes, a schematic for the structure of the $N_h$- layer network is shown in Figure 1, in which $N_h = 2$; $p_j$ $(j = 1, \dots, R)$ denotes the R inputs; $w_{ij}^m$ is the weight representing the connection to the $i^{\text{th}}$-neuron of the $m^{\text{th}}$-layer from the $j^{\text{th}}$-neuron of the $(m-1)^{\text{th}}$-layer, such that $m = 1, \dots, N_h$, $i = 1, \dots, S^m$ and $j = 1, \dots, S^{(m-1)}$, as well as $S^m$, which is the total number of neurons used in the $m^{\text{th}}$-layer; $b_j^m$ is the bias of the $j^{\text{th}}$-neuro of the $m^{\text{th}}$-layer; $n_i^m$ denotes the net input of the $i^{\text{th}}$-neuron of the $m^{\text{th}}$-layer; $n_i^m = \sum_{j=1}^{S^{(m-1)}} w_{ij}^m a_j^{(m-1)} + b_j^m$; $a_i^m (i = 1, \dots, S^m)$ is the $i^{\text{th}}$-output of the $m^{\text{th}}$-layer, and $a_i^m = f^m(n_i^m)$, where $f^m$ is the transfer function used in the $m^{\text{th}}$-layer, and $a_j^0 = p_j$. A symbol of $R$-$S^1$-$S^2$ in this work was thus used to represent a two-layer network algorithm with $R$ inputs, $S^1$ and $S^2$ neurons in the first and second layers, and $S^2$ outputs, in which the number of neurons used for the last layer was the same as the number of final outputs.

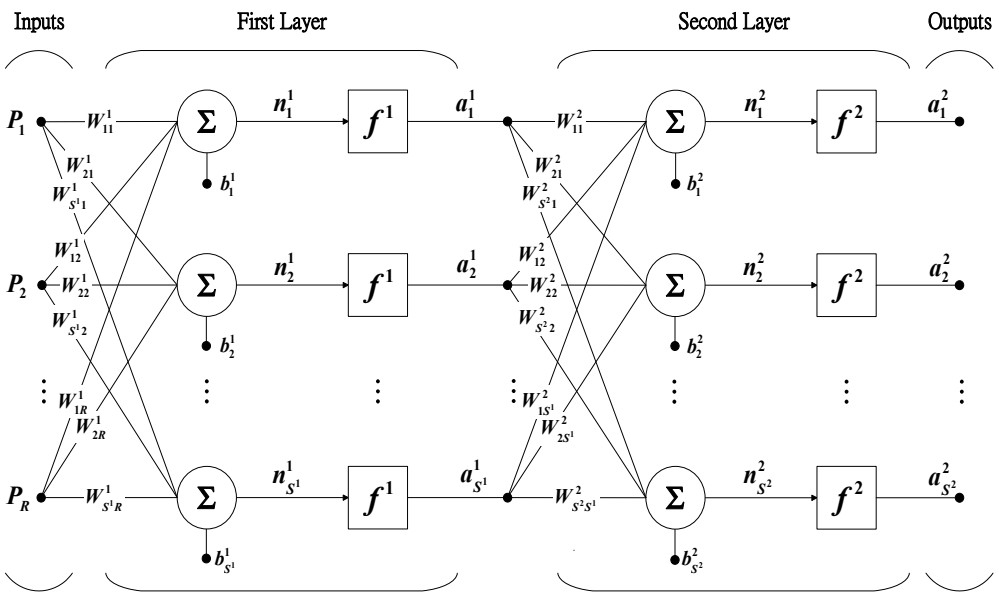

**Figure 1.** Schematic of the structure of a two-layer network.

For a multilayer network, the output of one layer became the input to the following layer, the operation of which as thus given as:

$$\mathbf{a}^{m+1} = \mathbf{f}^{m+1}\left(\mathbf{n}^{m+1}\right) = \mathbf{f}^{m+1}\left(\mathbf{W}^{m+1}\mathbf{a}^m + \mathbf{b}^{m+1}\right) \quad \text{for } m = N_h, \dots, 1, 0, \tag{20}$$

where $\mathbf{b}^{m+1} = \left\{ b_1^{m+1} \quad b_2^{m+1} \quad \cdots \quad b_{S^{m+1}}^{m+1} \right\}^T$; $\mathbf{n}^{m+1} = \left\{ n_1^{m+1} \quad n_2^{m+1} \quad \cdots \quad n_{S^{m+1}}^{m+1} \right\}^T$. In this work, the transfer function for the hidden layers of $m = 1, \dots, (N_h - 1)$ was selected as the log-sigmoid function, while the last output layer of m = $N_h$ was selected as the linear transfer function, such that $\mathbf{f}^{m+1} = \left\{ f_1^{m+1} \quad f_2^{m+1} \quad \cdots \quad f_{S^{m+1}}^{m+1} \right\}^T$, in which $f_i^{m+1} = 1/\left(1 + e^{-n_i^{m+1}}\right)$ and $f_i^{N_h} = n_i^{N_h}$.

In the first hidden layer, the neurons received external inputs as follows:

$$\mathbf{a}^0 = \mathbf{p}, \tag{21}$$

where $\mathbf{p} = \left\{ p_1 \quad p_2 \quad \cdots \quad p_R \right\}$, which provided the starting point for Equation (21).

The outputs of the neurons in the last layer were considered as the network outputs and defined as:

$$\mathbf{a} = \mathbf{a}^{N_h}. \tag{22}$$

*3.2. Backpropagation Process*

In order to operate the algorithm, there were appropriate input data sets and their corresponding target output sets, which were provided and given as:

$$\left(\mathbf{p}_1, \ \mathbf{t}_1\right), \left(\mathbf{p}_2, \ \mathbf{t}_2\right), \dots \left(\mathbf{p}_q, \ \mathbf{t}_q\right), \dots, \left(\mathbf{p}_Q, \ \mathbf{t}_Q\right). \tag{23}$$

The learning rule used in this ANN algorithm is called the least mean square algorithm, which means that the network parameters could be adjusted to minimize the mean square error as follows:

$$
\begin{aligned}
\text{Min} \quad \hat{F}(\mathbf{x}) &= \sum_{q=1}^{Q} (\mathbf{t}_q - \mathbf{a}_q)^T (\mathbf{t}_q - \mathbf{a}_q) \\
&= \sum_{q=1}^{Q} (\mathbf{e}_q)^T \mathbf{e}_q = \sum_{q=1}^{Q} \sum_{j=1}^{S^{N_h}} (e_{jq})^2 \\
&= \sum_{i=1}^{n_e} (v_i)^2,
\end{aligned} \tag{24}
$$

where $\mathbf{x} = \left\{ w_{11}^1, \cdots, w_{S^1 R}^1, b_1^1, \cdots, b_{S^1}^1, \cdots, w_{11}^{N_h}, \cdots, w_{S^{N_h} S^{N_h-1}}^{N_h}, b_1^{N_h}, \cdots, b_{S^{N_h}}^{N_h} \right\}$, in which the total number ($n_p$) for the parameters, including all the weights and biases in this MP BPNN algorithm, is $n_p = (R+1)S^1 + (S^1+1)S^2 + \cdots (S^{N_h-1}+1)S^{N_h}$. $\mathbf{v}^T = \{ v_1 \; v_2 \; \cdots \; v_{n_e} \} = \left\{ e_{11} \; e_{21} \; \cdots \; e_{S^{N_h} 1} \; e_{12} \; e_{22} \; \cdots \; e_{S^{N_h} 2} \; \cdots \; e_{S^{N_h} Q} \right\}$, in which the total number ($n_e$) of the square error terms is $n_e = Q \, S^{N_h}$.

In order to speed up the convergence rate, the Levenberg–Marquardt algorithm [26] was adopted for network training.

When the input $\mathbf{P}_q$ was fed into the network and the corresponding network output $\mathbf{a}_q^{N_h}$ was computed, the Levenberg–Marquardt BP algorithm could be initialized with:

$$
\overset{\sim}{\mathbf{S}}^{N_h} = \begin{bmatrix} \overset{\sim}{\mathbf{S}}_1^{N_h} & \overset{\sim}{\mathbf{S}}_2^{N_h} & \cdots & \overset{\sim}{\mathbf{S}}_Q^{N_h} \end{bmatrix} \tag{25}
$$

where $\overset{\sim}{\mathbf{S}}_q^{N_h} = -\dot{\mathbf{F}}^{N_h}\left(\mathbf{n}_q^{N_h}\right)$, and the dimension of $\dot{\mathbf{F}}^m(\mathbf{n}^m)$ is $S^m \mathrm{x} S^m$ and that of $\overset{\sim}{\mathbf{S}}^m$ is $S^m \times (Q \, S^m)$ for m = 1, ..., $N_h$.

The total Marquardt sensitivity matrices $\overset{\sim}{\mathbf{S}}^m$ could then be obtained in the following order: $m = N_h - 1, N_h - 2, \ldots, 1$, using the BP network, and this could be written as follows:

$$
\overset{\sim}{\mathbf{S}}^m = \begin{bmatrix} \overset{\sim}{\mathbf{S}}_1^m & \overset{\sim}{\mathbf{S}}_2^m & \cdots & \overset{\sim}{\mathbf{S}}_Q^m \end{bmatrix} \quad \text{for } m = N_h - 1, N_h - 2, \ldots, 1, \tag{26}
$$

where $\overset{\sim}{\mathbf{S}}_q^m = \dot{\mathbf{F}}^m\left(\mathbf{n}_q^m\right)\left(\mathbf{W}^{m+1}\right)^T \overset{\sim}{\mathbf{S}}_q^{m+1}$ for q = 1, ..., Q.

According to the Levenberg–Marquardt BP algorithm, all of the parameters could be modified using an iteration process as follows:

$$
\mathbf{X}^{(k+1)} = \mathbf{X}^{(k)} - \left[ \mathbf{J}^T\left(\mathbf{x}^{(k)}\right)\mathbf{J}\left(\mathbf{x}^{(k)}\right) + \mu^{(k)} \, \mathbf{I} \right]^{-1} \mathbf{J}^T\left(\mathbf{x}^{(k)}\right)\mathbf{v}\left(\mathbf{x}^{(k)}\right), \tag{27}
$$

where *k* denotes the number of iterations, and the initial value of $\mu$, i.e., $\mu^{(0)}$, was taken to be a significant value (e.g., $\mu^{(0)}$ = 10,000). If a step yielded a more significant value for F(**x**), then the step was repeated with $\mu^{(k)}$ multiplied by a factor $\vartheta > 1$ (e.g., $\vartheta$ = 2 in this work), and if a step yielded a smaller value for F(**x**), then the step was repeated with $\mu^{(k)}$ divided by a factor $\vartheta > 1$ (e.g., $\vartheta$ = 3 in this work). Eventually, the iteration process converged. In this work, the stop criteria of the iteration process were considered as follows:

(1) When the root of the mean square relative error $R_e$ was less than $10^{-5}$, i.e.,

$$
R_e = \sqrt{\sum_{q=1}^{Q} \left[ (\mathbf{t}_q - \mathbf{a}_q)^T (\mathbf{t}_q - \mathbf{a}_q) / \left(\mathbf{t}_q^T \mathbf{t}_q\right) \right] / (Q \, S^{N_h})} \leq 10^{-5} \tag{28}
$$

(2) When the number of iterations was greater than 20,000.

## 4. Illustrative Examples

### 4.1. Large Amplitude-Free Vibration Analysis Using the Mixed FE Method

In this section, the authors investigate the large amplitude-free vibration behavior of homogeneous isotropic and FG isotropic beams with combinations of simply supported, free, and clamped boundary conditions using the mixed FE method.

Table 1 shows the results of the convergence study for the mixed FE solutions of the nonlinear-to-linear natural frequency ratios ($\omega_{nl}/\omega_{linear}$) of homogeneous isotropic beams under S-S boundary conditions. This issue has also been studied using various FE methods, such as the Lagrange-type FE method [43] and the Galerkin FE method [44]. The accuracy and convergence rate of the RMVT-based FE methods with different orders used to expand the primary variables in the element domain could thus be validated by comparing their solutions with those available in the literature.

**Table 1.** Results of the convergence study for the mixed FE solutions of the nonlinear-to-linear natural frequency ratios of isotropic beams under S-S boundary conditions.

| Theories | $w_{\mathbf{max}}/\sqrt{I/A}$ | | | |
|---|---|---|---|---|
| | **1.0** | **2.0** | **3.0** | **4.0** |
| Current four linear elements | 1.1196 | 1.4193 | 1.8117 | 2.2488 |
| Current eight linear elements | 1.1187 | 1.4164 | 1.8065 | 2.2412 |
| Current 16 linear elements | 1.1187 | 1.4163 | 1.8062 | 2.2407 |
| Current four quadratic elements | 1.1173 | 1.4124 | 1.8006 | 2.2342 |
| Current eight quadratic elements | 1.1183 | 1.4155 | 1.8053 | 2.2398 |
| Current 16 quadratic elements | 1.1186 | 1.4161 | 1.8061 | 2.2406 |
| Current four cubic elements | 1.1187 | 1.4164 | 1.8064 | 2.2410 |
| Current eight cubic elements | 1.1187 | 1.4162 | 1.8062 | 2.2407 |
| Sarma and Varadan [43] | 1.1180 | 1.4142 | 1.8028 | 2.2361 |
| Bhashyam and Prathap [44] | 1.1180 | 1.4141 | 1.8027 | 2.2359 |

The geometric parameters of the beams were $L/h$ = 20 and $h$ = 0.1 m. The material properties were $E$ = 322.3 GPa, and $v$ = 0.24. A maximum dimensionless amplitude was defined as obtaining the nonlinear natural frequencies of the beams.

It can be seen in Table 1 that the RMVT-based (or mixed) FE solutions converged rapidly and that convergent solutions were obtained when the 16 linear, eight quadratic or four cubic elements were used. Moreover, the convergent solutions were in excellent agreement with those obtained using various PVD-based FE methods [43,44].

Table 2 compares the mixed FE solutions for the natural frequency ratios of FG beams under C-C boundary conditions with the solutions reported in the literature, in which 16 cubic elements were used. In addition, mixed FE solutions for the S-S, C-S, and C-F boundary conditions were also presented. The FG beam was made of a ceramic-metal two-phase material, comprised Silicon nitride Si3N4 (ceramic) and a stainless steel SuS304 (metal), and the material properties of the Si3N4 and SuS304 materials were $E_c$ = 322.3 GPa, $v_c$ = 0.24, and $\rho_c$ = 2370 Kg/m$^3$, as well as $E_m$ = 207.8 GPa, $v_m$ = 0.3178, and $\rho_m$ = 8166 Kg/m$^3$, in which the subscripts c and m denoted the ceramic and metal materials, respectively. The material properties of the FG beam were assumed to obey the power-law distributions through the thickness direction according to the volume fractions of the constituents, which were $\Gamma_c(z) = [1/2 + (z/h)]^{\kappa_p}$ and $\Gamma_m(z) = 1 - \Gamma_c(z)$, in which $\kappa_p$ represents the material-property gradient index, and the effective material properties of this were estimated using the rule of mixture, as follows:

$$P(z) = P_c \Gamma_c(z) + P_m \Gamma_m(z) \tag{29}$$

where $P$ represents $E$, $v$, and $\rho$.

**Table 2.** Comparisons between the mixed FE solutions of the nonlinear-to-linear natural frequency ratios ($\omega_{nl}/\omega_{linear}$) of FG beams under C-C boundary conditions and the solutions reported in the literature, in which the mixed FE solutions for the S-S, C-S, and C-F boundary conditions are also presented.

| Boundary Conditions | $\kappa_p$ | Theories | $\overline{W}_{max} = W_{max}/\sqrt{I/A}$ | | | |
|---|---|---|---|---|---|---|
| | | | **1** | **2** | **3** | **4** |
| C-C | 0.3 | Mixed FE method | 1.0305 | 1.1164 | 1.2445 | 1.4017 |
| | | Elmaguiri et al. [45] | 1.0227 | 1.0875 | 1.1869 | 1.3121 |
| | | Ke et al. [46] | 1.0220 | 1.0852 | 1.1831 | 1.3079 |
| | 1 | Mixed FE method | 1.0304 | 1.1158 | 1.2434 | 1.4000 |
| | | Elmaguiri et al. [45] | 1.0225 | 1.0871 | 1.1860 | 1.3106 |
| | | Ke et al. [46] | 1.0025 | 1.0873 | 1.1874 | 1.3149 |
| | 2 | Mixed FE method | 1.0296 | 1.1128 | 1.2374 | 1.3906 |
| | | Elmaguiri et al. [45] | 1.0215 | 1.0832 | 1.1780 | 1.2980 |
| | | Ke et al. [46] | 1.0232 | 1.0900 | 1.1929 | 1.3237 |
| S-S | 0.3 | Mixed FE method | 1.1492 | 1.4666 | 1.8681 | 2.3102 |
| | 1 | Mixed FE method | 1.1704 | 1.4964 | 1.8993 | 2.3397 |
| | 2 | Mixed FE method | 1.1695 | 1.4899 | 1.8855 | 2.3182 |
| C-S | 0.3 | Mixed FE method | 1.0696 | 1.2379 | 1.4667 | 1.7299 |
| | 1 | Mixed FE method | 1.0764 | 1.2481 | 1.4779 | 1.7409 |
| | 2 | Mixed FE method | 1.0755 | 1.2438 | 1.4691 | 1.7272 |
| C-F | 0.3 | Mixed FE method | 1.00004 | 0.99998 | 0.99979 | 0.99948 |
| | 1 | Mixed FE method | 1.00008 | 1.00004 | 0.99989 | 0.99961 |
| | 2 | Mixed FE method | 1.00006 | 1.00002 | 0.99985 | 0.99956 |

It can be seen in Table 2 that the nonlinear-to-linear natural frequency ratios increased when the maximum dimensionless amplitude became greater for C-C, C-S, and S-S boundary conditions. Again, it can be seen in Table 2 that for the C-C boundary conditions, the mixed FE solutions closely agreed with the solutions obtained by Elmaguiri et al. [45] and Ke et al. [46] with Euler-Bernoulli's beam theory accounting for the VKGN effect. These results also show that the magnitudes of the nonlinear-to-linear natural frequency ratio at increments for different boundary conditions were arranged in descending order: S-S boundary conditions > C-S boundary conditions > C-C boundary conditions.

On the other hand, for C-F boundary conditions, the nonlinear-to-linear natural frequency ratios slightly decreased when the maximum dimensionless amplitude became greater. This was because the geometrical nonlinearity effect and the boundary constraints resulted in an axial tensional force for the S-S, C-S, and C-C boundary conditions, enhancing the overall stiffness of the FG beams. In contrast, this resulted in a very small compressive force for the C-F boundary conditions, which slightly weakened the overall stiffness of the FG beams. The results also showed that the influence of the geometrical nonlinearity effect on the natural frequencies of the FG beams for different boundary conditions was arranged in descending order: S-S > C-S > C-C > C-F boundary conditions.

The modal shapes of the FG beam under different boundary conditions are shown in Figure 2, in which the maximum dimensionless amplitude and the material-property gradient index were taken to be $\overline{w}_{max} = w_{max}/\sqrt{I/A} = 1$ and $\kappa_p = 0.3$, respectively.

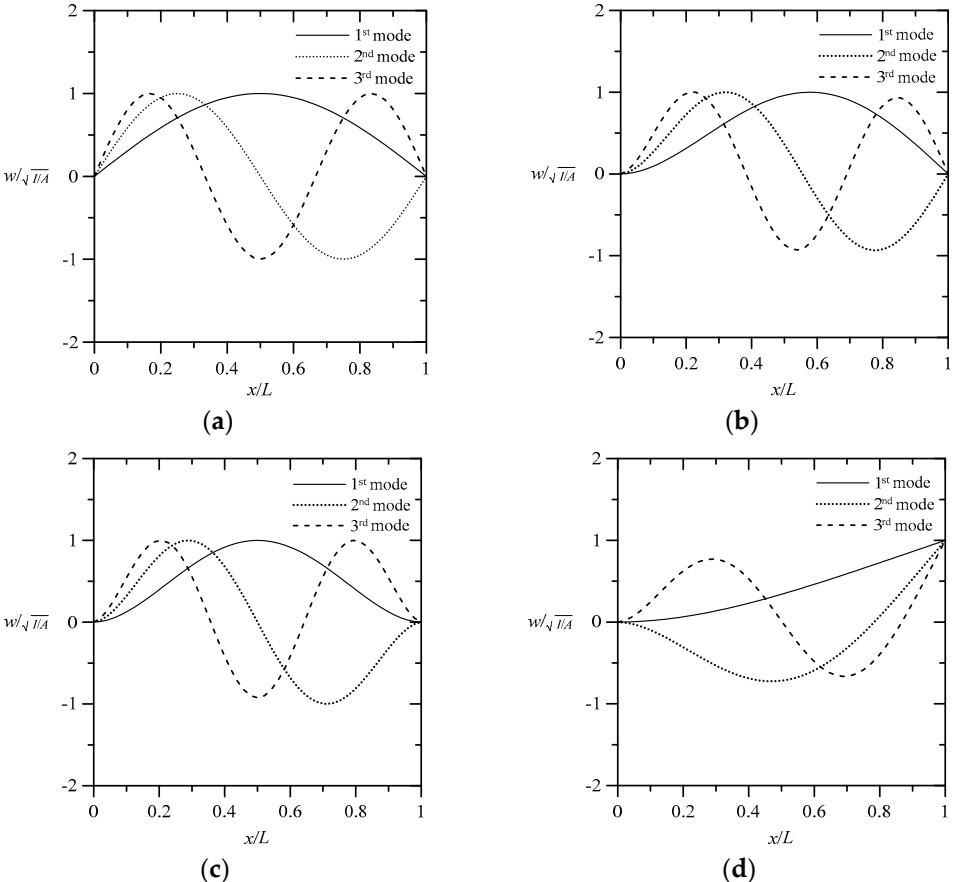

**Figure 2.** The 1st, 2nd, and 3rd mode shapes of the FG beams under (**a**) S−S, (**b**) C−S, (**c**) C−C, and (**d**) C−F boundary conditions.

*4.2. Large Amplitude–Free Vibration Analysis Using the Developed MP BPNN*

In this section, the large amplitude free vibration analysis of the FG rectangular beams mentioned above was undertaken again using the developed MP BPNN algorithm, which was trained using the mixed FE solutions. The boundary conditions of the FG beams were considered to be clamped-clamped edge conditions. The cross-section of the FG beam was 10 cm × 10 cm, and their material properties were considered the same as those used in Table 2. The dimensionless nonlinear frequency parameter was defined as $\overline{\omega}_{nl} = \omega_{nl} \sqrt{\rho_m L^2 / E_m}$.

After training, a set of optimal parameters, including the weight numbers $w_{ij}^m$ and biases $b_j^m$, where $i = 1, \ldots, S^m$, $j = 1, \ldots, S^{(m-1)}$, and $m = 1, \ldots, N_h$, could be obtained, and the trained MP BPNN could replace the mixed FE method to predict the nonlinear natural frequencies of the FG beams significantly faster compared to the case with the mixed FE method. In the training of this MP BPNN, $R$ was taken as three and represented the length-to-thickness ratio ($L/h$), the material-property gradient index ($\kappa_p$), and the maximum dimensionless amplitude ($w_{max}/ \left( \sqrt{I/A} \right)$), in which $L/h = 10^{1+0.05i}$, $i = 0, \ldots, 20$; $\kappa_p = 10^{-2+0.2i}$, $i = 0, \ldots, 20$; $w_{max}/ \left( \sqrt{I/A} \right) = 1 + i$, $i = 0, \ldots, 3$, such that the total number of the training data sets was 21 × 21 × 4 = 1764. The number of outputs was taken as two, which were the first and second lowest natural frequencies of the FG beams. Thus, there were 1764 training data sets for the first lowest and second lowest nonlinear natural frequencies of the FG beams, which were obtained using the mixed FE methods and used to train the $n_p$ parameters for each MP BPNN, in which $n_p = (R+1)S^1 + (S^1+1)S^2 + \cdots (S^{N_h-1}+1)S^{N_h}$, the initial values of which were randomly generated between −1 and 1. There were 100 sets of the initial values of the $n_p$ parameters used for each MP BPNN algorithm in this analysis, for which the first three best

results are shown in Table 3 and were estimated using the $R_e$ of the outputs compared to the other 200 selected testing data.

**Table 3.** General information regarding the developed MP BPNNs and their performance.

| No. Neurons of Each Hidden Layer | No. Hidden Layers (i.e., $(N_h - 1)$) | | | | | | | | | | | |
|---|---|---|---|---|---|---|---|---|---|---|---|---|
| | **1** | | | **2** | | | **3** | | | **4** | | |
| | No. Parameters | Training Time (s) | $R_e$ of Outputs | No. Parameters | Training Time (s) | $R_e$ of Outputs | No. Parameters | Training Time (s) | $R_e$ of Outputs | No. Parameters | Training Time (s) | $R_e$ of Outputs |
| 2 | 14 | 40.99 | 0.2328% | 20 | 116.50 | 0.1913% | 26 | 83.46 | 0.0341% | 32 | 314.25 | 0.0296% |
| | | 31.98 | 0.2331% | | 170.48 | 0.1917% | | 52.81 | 0.1315% | | 525.97 | 0.0569% |
| | | 32.50 | 0.2331% | | 103.02 | 0.2206% | | 56.06 | 0.1426% | | 107.98 | 0.0572% |
| 4 | 26 | 45.13 | 0.1393% | 46 | 70.94 | 0.0048% | 66 | 435.87 | 0.0008% | | | |
| | | 25.45 | 0.1421% | | 38.68 | 0.0167% | | 188.26 | 0.0043% | | | |
| | | 35.50 | 0.2676% | | 124.84 | 0.0168% | | 78.46 | 0.0128% | | | |
| 6 | 38 | 163.32 | 0.0237% | 80 | 173.84 | 0.0006% | | | | | | |
| | | 55.70 | 0.0338% | | 252.95 | 0.0007% | | | | | | |
| | | 55.11 | 0.0467% | | 533.02 | 0.0008% | | | | | | |
| 8 | 50 | 82.67 | 0.0228% | | | | | | | | | |
| | | 86.69 | 0.0266% | | | | | | | | | |
| | | 122.42 | 0.0359% | | | | | | | | | |

In this section, the large amplitude free vibration analysis of FG rectangular beams mentioned above was undertaken again using the developed MP BPNN algorithm, which was trained using the mixed FE solutions. The FG beams' boundary conditions were considered clamped-clamped edge conditions. The cross-section of the FG beam was 10 cm × 10 cm, and their material properties were considered the same as those used in Table 2. The dimensionless nonlinear frequency parameter was defined as $\overline{\omega}_{nl} = \omega_{nl} \sqrt{\rho_m L^2 / E_m}$.

After training, a set of optimal parameters, including the weight numbers $w_{ij}^m$ and biases $b_j^m$, where $i = 1, \ldots, S^m$, $j = 1, \ldots, S^{(m-1)}$, and $m = 1, \ldots, N_h$, could be obtained, and the trained MP BPNN replaced the mixed FE method to predict the nonlinear natural frequencies of the FG beams significantly faster compared to the case with the mixed FE method. In the training of this MP BPNN, $R$ was taken as three, and represented the length-to-thickness ratio $(L/h)$, the material-property gradient index $(\kappa_p)$, and the maximum dimensionless amplitude $(w_{\max} / (\sqrt{I/A}))$, in which $L/h = 10^{1+0.05i}$, $i = 0, \ldots, 20$; $\kappa_p = 10^{-2+0.2i}$, $i = 0, \ldots, 20$; $w_{\max} / (\sqrt{I/A}) = 1 + i$, $i = 0, \ldots, 3$, such that the total number of the training data sets was $21 \times 21 \times 4 = 1764$. The number of outputs was taken as two, which were the first and second lowest natural frequencies of the FG beams. Thus, there were 1764 training data sets for the first lowest and second lowest nonlinear natural frequencies of the FG beams obtained using the mixed FE methods, and these were used to train the $n_p$ parameters for each MP BPNN, in which $n_p = (R+1)S^1 + (S^1+1)S^2 + \cdots (S^{N_h-1}+1)S^{N_h}$, the initial values of which were randomly generated between $-1$ and 1. There were 100 sets of the initial values of the $n_p$ parameters used for each MP BPNN algorithm in this analysis, for which the first three best results are shown in Table 3; these were estimated using the $R_e$ of outputs as compared with another 200 selected testing data.

It can be seen in Table 3 that the 3-6-6-2 and 3-4-4-4-2 BPNN algorithms yielded optimal results, for which the values of $R_e$ were 0.0006% and 0.0008%, respectively. The 3-6-6-2 BPNN algorithm was thus used to predict the first lowest and second lowest nonlinear natural frequencies of the FG beams, as shown in Figures 3 and 4. Furthermore, it can be seen in each curve of Figures 3 and 4 that the solutions obtained using the mixed FE method, drawn using the solid lines and the developed MP BPNN algorithm, and drawn using the discrete symbols closely agreed with each other for a randomly selected group of 50 sets of input data.

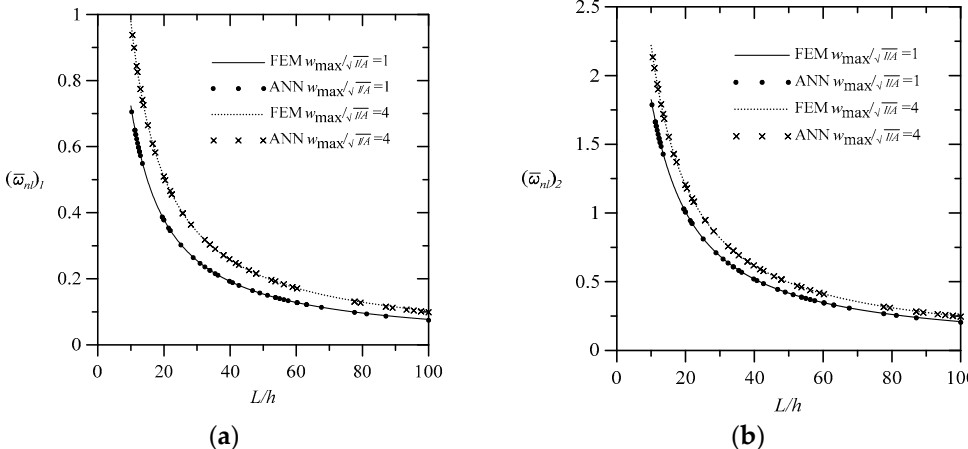

**Figure 3.** Variations in the nonlinear frequency parameters of FG beams with the aspect ratio using the mixed FE method and the developed MP BPNN algorithm for $w_{max}/\sqrt{I/A} = 1$ and 4, and $\kappa_p = 4$; (**a**) The first lowest frequency parameters, (**b**) The second lowest frequency parameter.

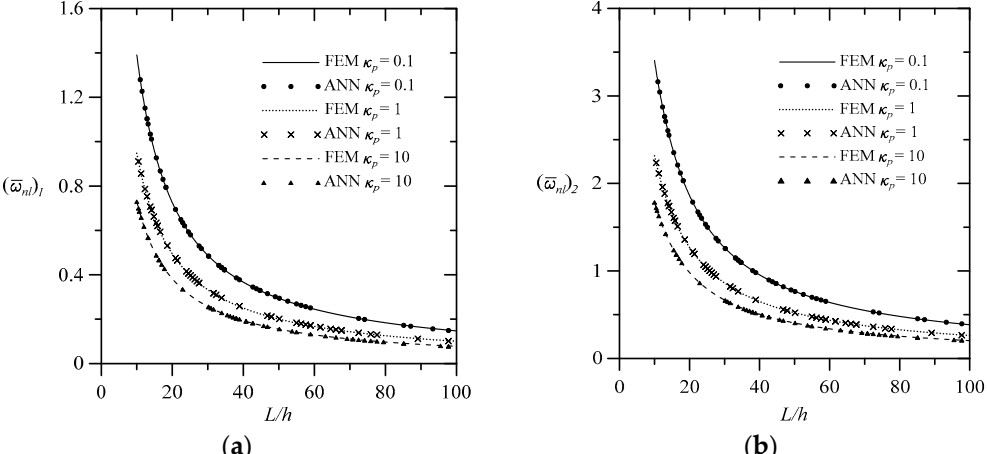

**Figure 4.** Variations in the nonlinear frequency parameters of FG beams with the aspect ratio using the mixed FE method and the proposed MP BPNN algorithm for $\kappa_p = 0.1$, 1, and 10, and $w_{max}/\sqrt{I/A} = 2$; (**a**) The first lowest frequency parameters, (**b**) The second lowest frequency parameters.

Table 4 compares the computer execution time required for the developed MP BPNN and the mixed FE method when randomly selecting 50 input data sets, as shown in Figures 3 and 4. It was shown that the central processing unit (CPU) time required for the mixed FE method was about 2800 times greater than that needed for the developed MP BPNN algorithm. Thus, the trained MP BPNN algorithm was superior to the mixed FE method because it was less time-consuming.

The results also showed that the nonlinear frequency parameters decreased when the length-to-thickness ratio increased. The nonlinear frequency parameters depended on the maximum dimensionless amplitude, and these values increased when the maximum dimensionless amplitude was greater.

**Table 4.** A comparison of the CPU time required for the proposed MP BPNN and the mixed FEM when randomly selecting a group of 50 input data sets is shown in Figures 2 and 3.

| $\left[\kappa_p,\ w_{\max}/\sqrt{I/A}\right]$ | Time (s) | |
|:---:|:---:|:---:|
| | **ANN** | **FEM** |
| [4, 1] | 0.0027 | 7.1167 |
| [4, 4] | 0.0025 | 7.5926 |
| [0.1, 2] | 0.0026 | 7.4780 |
| [1, 2] | 0.0026 | 7.0521 |
| CPU: Central processing unit. | | |

## 5. Conclusions

In this article, based on the RMVT and TBT, a mixed FE method was developed for the nonlinear free vibration analysis of FG beams under the combinations of simply supported, free, and clamped edge conditions. Implementing the mixed FE method shows that its convergent solutions were obtained when 16 linear, eight quadratic, or four cubic elements were used and that these convergent solutions were in excellent agreement with the accuracy solutions reported in the literature. These results also indicated that the nonlinear natural frequency-to-linear natural frequency ratios increased when the maximum dimensionless amplitude increased. On the other hand, for C-F boundary conditions, the nonlinear-to-linear natural frequency ratios decreased when the maximum dimensionless amplitude became greater. The influence of the geometrical nonlinearity effect on the natural frequencies of the FG beams for different boundary conditions was arranged in descending order: S-S > C-S > C-C > C-F boundary conditions.

An MP BPNN algorithm was also developed, in which the Levenberg–Marquardt algorithm was used to speed up MP BPNN's convergence rate. After training using the mixed FE solutions, it was shown that the trained MP BPNN algorithm was superior to the mixed FE method because it was approximately 1/2800 times less time-consuming. It is worth mentioning that the above comparison did not account for the cost of generating FEM data and training the BPNN. It is expected that the developed MP BPNN algorithm could be extended to research subjects, including inverse problems, optimal design, structural control, etc., when the traditional analytical or numerical methods used in such studies are very time-consuming. Therefore, applying the MP BPNN to the research mentioned above is ongoing.

**Author Contributions:** Conceptualization, C.-P.W.; methodology, C.-P.W.; software, S.-T.Y. and J.-H.L.; validation, C.-P.W., S.-T.Y. and J.-H.L.; formal analysis, C.-P.W.; investigation, C.-P.W.; resources, C.-P.W.; data curation, S.-T.Y. and J.-H.L.; writing—original draft preparation, C.-P.W.; writing—review and editing, C.-P.W.; visualization, C.-P.W.; supervision, C.-P.W. All authors have read and agreed to the published version of the manuscript.

**Funding:** This research received no external funding.

**Institutional Review Board Statement:** Not applicable.

**Informed Consent Statement:** Not applicable.

**Data Availability Statement:** The data supporting reported results can be found at https://drive.google.com/drive/folders/1TmbVHAdo6sZNMGytFrmZPrLPL4_Ye4aN?usp=share_link (accessed on 1 May 2023).

**Conflicts of Interest:** The authors declare no conflict of interest.

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
