# Peer review of "A Nonlinear Free Vibration Analysis of Functionally Graded Beams Using a Mixed Finite Element Method and a Comparative Artificial Neural Network"

_jcs, doi:10.3390/jcs7060229_

Round 1

Reviewer 1 Report

In this research authors have used the first order shear deformation beam theory to analyse the small and large amplitude vibrations of functionally graded beams. Properties of the beam are assumed to be graded through the thickness direction. To consider the geometrically nonlinear effects,
von Karman type of geometrical non-linearity is taken into account. The energies of the beam are obtained and a finite element solution method is applied to discrete the problem.
it also should be noted that different type of edge supports such as clamped, simply supported and free are assumed for the beams. The obtained equations are then solved using a direct iterative algorithm to reach the amplitude-vibration curves. A multilayer perceptron (MP) back propagation neural network (BPNN) is also established to estimate the nonlinear free vibration behavior of beam. The results of this study are well-compared with the available data in the open literature and after that novel numerical results are given. The provided manuscript is interesting and may be considered for publication  when the following comments are correctly answered
1) It is well-known that using the FSDT may face the shear locking phenomenon in finite element programming. How authors have solved this issue?
2) Finite element solution in nonlinear formulation may face the membrane locking phenomenon. Have you consider it in this research?
3) What is the physical interpretation of the fact that C-F beam represents the softening behavior while other type of edge support exhibit hardening response
4) FGM structures are asymmetric with respect to the mid-surface. So the negative and positive deflections are un-equal in large-amplitude vibrations.
How authors have considered such feature?
5) In Capation of table (2) authors have mentioned that a comparison is done. However there is no comparison in this table
6) In nonlinear free vibration problems generally backbone curved should be plotted. However authors have ignored to plot such curves. Why?
7) The problem of large amplitude free vibration of FGM beams may be solved under various effects. What is the need to use NN?
8) Enrich the literature review on the nonlinear free vibration of beam structures by considering more works such as
[Acta Mechanica 227 (7), 1869-1884, 2016]
[Iranian Journal of Science and Technology, Transactions of Mechanical Engineering, 45, 611-630, 2021]

Author Response

Dear Reviewers:

Please see the attached file, which replied to all of the comments raised by three reviewers. Thank you for your helpful comments and great patience in reviewing the manuscript.

Best regards,

Chih-Ping Wu

Reviewer 2 Report

The authors presents an effective numerical approach which combines mixed FEM and neural networks for vibration analysis of FG beams. To improve their work, the authors should report also the comparisons with the other numerical methods mentioned at line 323. Moreover, the final claim of alleged superiority of their approach over standard FEM is not well supported by results: the authors do not consider the cost for generating FEM data and for training the BPNN. 

I have several comments and modifications requests, which are reported in the attached pdf file. Please, to see them, use the "Comment" section under Adobe Acrobat.  

The paper is well written. Only few amendments are indicated in the attached file. 

Author Response

Dear Reviewers:

Please see the attached file, which replied to all of the comments raised by three reviewers. Thank you for your helpful comments and great patience in reviewing the manuscript.

Best regards,

Chih-Pig Wu

Reviewer 3 Report

Review on the Manuscript:

Paper ID: jcs-2415601

Title: A nonlinear free vibration analysis of functionally graded beams using a mixed finite element method and a comparative artificial neural network

In this work, the authors derive a weak-form formulation of and the Reissner mixed variation theorem and Timoshenko beam theory, for a large amplitude-free vibration analysis of functionally graded beams with combinations of simply-supported and clamped edge conditions, in which the von Kármán geometrical nonlinearity effect is considered.

The contents are relevant to this journal. I recommend its publication but before publication, I suggest following revision:

1.      The novelty of the work should be highlighted to real physics phenomena in ‎the introduction.‎

2.      The authors should try to give advantageous of using of their method compared to others.

3.      The obtained findings of this work should be compared to experimental results or at-least with ‎other published results in the literature.‎

4.      On page 5, line 208, the parameter m1 is missing.

5.      On page 5, line 210 does not make sense, or something is missing after line 209.

6.      On page 11, Fig. 2., missing  (b) C-S, (c) C-C, and (d) C-F.

If the authors take into account all these corrections, then this manuscript deserves to be published.

19.05.2023

Author Response

Dear Reviewers:

Please see the attached file, which replied to all of the comments raised by three reviewers. Thank you for your helpful comments and great patience in reviewing the manuscript.

Best regards,

Chih-Ping 

Round 2

Reviewer 1 Report

My comments are totally considered by authors and the manuscript may be accepted in its current form

Reviewer 2 Report

All comments and suggestions have been properly addressed by the authors. 

Reviewer 3 Report

Review on the Manuscript:

Paper ID: jcs-2415601

Title: A nonlinear free vibration analysis of functionally graded beams using a mixed finite element method and a comparative artificial neural network

    In my opinion, taking into account the author's answers and corrections, I recommend the acceptance of the manuscript for publication. However, the final decision is up to the editor-in-chief.

29.05.2023